# Polymorphism in the Chloroplast ATP Synthase Beta-Subunit Is Associated with a Maternally Inherited Enhanced Cold Recovery in Cucumber

**DOI:** 10.3390/plants10061092

**Published:** 2021-05-29

**Authors:** Madeline W. Oravec, Michael J. Havey

**Affiliations:** 1Department of Horticulture, University of Wisconsin, 1575 Linden Drive, Madison, WI 53706, USA; oravec@umn.edu; 2Department of Plant and Microbial Biology, University of Minnesota, 1479 Gortner Avenue, St. Paul, MN 55108, USA; 3USDA-ARS and Department of Horticulture, University of Wisconsin, 1575 Linden Drive, Madison, WI 53706, USA

**Keywords:** chilling stress, gene regulation, abiotic stress, cold tolerance, plastid genetics, breeding, cucurbits, *Cucumis sativus*

## Abstract

Cucumber (*Cucumis sativus* L.) is a warm-season crop that is sensitive to chilling temperatures and a maternally inherited cold tolerance exists in the heirloom cultivar ‘Chipper’ (CH). Because the organelles of cucumber show differential transmission (maternal for chloroplast and paternal for mitochondrion), this cold tolerance is hypothesized to be chloroplast-associated. The goal of this research was to characterize the cold tolerant phenotype from CH and determine its genetic basis. Doubled haploid (DH) lines were produced from CH and cold susceptible cucumbers, reciprocal hybrids with identical nuclear genotypes were produced, and plants were subjected to cold treatments under lights at 4 °C for 5.5 h. Hybrid plants with CH as the maternal parent had significantly higher fresh and dry weights 14 days after cold treatment compared to the reciprocal hybrid, revealing an enhanced cold recovery phenotype maternally conferred by CH. Results from analyses of the nuclear transcriptome and reactive oxygen species (ROS) between reciprocal hybrids were consistent with the cold recovery phenotype. Sequencing of the chloroplast genome and transcriptome of the DH parents and reciprocal hybrids, respectively, revealed one maternally transmitted non-synonymous single nucleotide polymorphism (SNP) in the chloroplast F_1_F_O_-ATP synthase (CF_1_F_O_-ATPase) beta-subunit gene (atpB) of CH which confers an amino acid change from threonine to arginine. Protein modeling revealed that this change is located at the interface of the alpha- and beta-subunits in the CF_1_F_O_-ATPase complex. Polymorphisms in the CF_1_F_O_-ATPase complex have been associated with stress tolerances in other plants, and selection for or creation of polymorphic beta-subunit proteins by chloroplast transformation or gene editing could condition improved recovery from cold stress in plants.

## 1. Introduction

Cucumber (*Cucumis sativus* L.) is a cold-sensitive crop that can be damaged by temperatures between 1 and 12 °C over relatively short periods of time [1,2]. Approaches such as spraying with hydrogen peroxide or uniconazole have been used to prepare crops for cold, but spraying is labor intensive and weather dependent [3,4]. Plastic tunnels can buffer against extreme temperatures, but entail high infrastructural costs and can increase pest pressure problems due to micro-environmental conditions and difficulty of spraying [5]. Delayed planting can reduce the risk of cold damage; however, crops must then remain in the field later into the summer when disease and pest pressures become high.

Natural variation for cold tolerance exists in plants and quantitative trait loci (QTL) conditioning cold tolerant phenotypes have been identified across crop species [6,7,8,9,10]. For example, the COLD1 (chilling tolerant divergence 1) gene has been implicated in cold tolerance in rice (*Oryza sativa*) and homologs associated with this stress tolerance have been identified in maize (*Zea mays*), sorghum (*Sorghum bicolor*), and other monocot species [11,12,13]. Variation in the CTB4a (cold tolerance at booting stage) gene in rice confers enhanced seed set and yield under cold conditions associated with increased adenosine triphosphate (ATP) supply [14]. Cold tolerance in the cucumber population NC-76 was assessed by leaf damage after cold exposure and conditioned by a dominant allele at the *Ch* locus [15]. While various studies in Arabidopsis (*Arabidopsis thaliana*) have demonstrated a central role for CBF/DREB1 (C-repeat binding factor/dehydration-responsive element binding) transcription factors in regulating COR (cold-responsive) genes during cold acclimation, the role of this pathway in chilling response is still not fully characterized [16,17]. Over-expression of the SlPIF4 (phytochrome-interaction transcription 4) gene in tomato (*Solanum lycopersicum*) increased cold tolerance through direct interaction and activation of CBF genes and altered plant hormone signaling [18]. Transgenic approaches using CBFs and other cold response genes have successfully improved cold tolerance in both chilling tolerant and sensitive species, but constitutive overexpression can also confer reduced growth and productivity [19].

A maternally inherited cold tolerance has been reported in the heirloom cucumber cultivar ‘Chipper’ (CH), and is expressed as reduced leaf damage from temperatures down to 4 °C for periods up to 5.5 h [2]. Maternal inheritance of this trait suggests that this cold tolerance is conditioned by the chloroplast genome, because the chloroplast and mitochondrial DNAs of cucumber are maternally versus paternally transmitted, respectively [20]. PCR amplicons from the chloroplast DNA of CH and cold susceptible cucumbers were sequenced and revealed three single nucleotide polymorphisms (SNPs) associated with cold tolerance [4]. One SNP was located in a non-coding region between the genes trnK (tRNA-Lys) and rps16 (ribosomal protein s16). The other two SNPs were a synonymous SNP in the ycf1 gene and a non-synonymous SNP in the chloroplast F_1_F_O_-ATP synthase (CF_1_F_O_-ATPase) beta-subunit gene (atpB) [4]. However, chloroplast transcripts can undergo RNA editing [21], and the possibility exists that editing may result in no change in the protein product [22,23,24]. Along with this, silent or synonymous mutations downstream of the initiation codon have been suggested to affect translation in chloroplasts [25]. Further investigation of the chloroplast transcriptome is therefore necessary to confirm any causative potential of these chloroplast DNA (cpDNA) polymorphisms.

The roles of organelles in abiotic stress signaling and tolerance continue to be recognized and reviewed in the literature, but are not yet fully elucidated [26,27,28,29,30,31]. Chloroplasts are early sensors of cold stress and retrograde signaling from the chloroplast initiates nuclear stress response pathways [28,29]. Reactive oxygen species (ROS) produced in the chloroplast under stresses, such as high light and low temperatures, can damage leaf tissues and photosynthesis, but also play a vital role in retrograde signaling to alter nuclear gene expression [32,33]. Overall, the central role of organelles in stress signaling is well established, and studying organellar genetics and properties should be valuable toward development of stress tolerant plants.

Cucumber is a unique model plant for studying organellar mutations and organellar-nuclear interactions because the three genomes are differentially transmitted, that is, the nuclear, mitochondrial, and chloroplast genomes are inherited biparentally, paternally, and maternally, respectively [20,34]. This allows for simple assignment of polymorphisms and phenotypes to specific genomes by reciprocal crossing. Along with this, the cucumber chloroplast, mitochondrial, and nuclear genomes have been sequenced, assembled, and annotated in multiple genetic backgrounds [4,35,36,37,38,39,40,41]. The goals of this research were to examine the phenotypic, genetic, and physiological characteristics of the cold tolerance from CH and reveal insights on mechanisms of stress tolerance in plants.

## 2. Results

### 2.1. CH Shows a Maternally Transmitted, Cold Recovery Phenotype

Growth of doubled haploids (DHs) of the cold tolerant cucumber ‘Chipper’ (CH) was compared to the two cold susceptible cucumber, ‘Straight 8’ (ST8) and ‘Marketmore 76’ (MM), after cold treatment and in control conditions. Plants of cold treated DH CH had significantly (*p* < 0.05) higher fresh and dry weights at 14 days after cold treatment compared to cold treated DHs ST8 and MM (Figure 1). No significant differences in fresh or dry weights were observed among DHs CH, ST8, and MM across all timepoints when plants were not cold treated (Figure 1). Notably, few plants from DH ST8 survived to 14 days after cold treatment. These results indicate that DH CH has enhanced recovery and regrowth after cold treatment compared to cold susceptible DHs ST8 and MM.

Reciprocal hybrids were developed using cold tolerant DH CH as the maternal or paternal parent in combination with the two cold susceptible DHs, ST8 and MM, and growth was compared across hybrids. For all hybrids, the maternal parent is listed first (i.e., CHxMM indicates that CH was the maternal parent and MM was the paternal parent). No significant differences in fresh or dry weight were observed across all timepoints between reciprocal hybrids from control conditions (Figure 2). When reciprocal hybrids of CH and ST8 were cold treated, the hybrid with CH as the maternal parent (CHxST8) had significantly (*p* < 0.05) higher fresh weight 14 days after cold treatment compared to its reciprocal (ST8xCH) hybrid (Figure 2). Dry weight of CHxST8 was significantly (*p* < 0.05) higher at both 7 and 14 days after cold treatment compared to ST8xCH (Figure 2). A similar response was observed for reciprocal hybrids between CH and MM, where CHxMM had significantly (*p* < 0.05) higher fresh and dry weight 14 days after cold treatment compared to MMxCH (Figure 2). However, few MMxCH plants survived to 14 days after cold treatment.

Visible leaf damage was apparent on the young leaves of all cold treated plants in the form of bleaching, necrosis, or leaf cupping (Appendix A). Though leaf damage was apparent across DHs and hybrids, the success and timing of recovery and regrowth varied across genotypes. Visible stunting or plant death was apparent in cold treated hybrids with CH as the paternal parent (ST8xCH and MMxCH) compared to their reciprocals, but no difference in growth were apparent under normal growing conditions (Appendix A). While no significant differences in growth patterns were observed for DHs compared to CH and among reciprocal hybrids in control conditions, significant differences (*p* < 0.05) in regrowth were observed after cold treatment. Results indicate that CH has an enhanced cold recovery phenotype that is maternally transmitted, but does not impart reduced growth under normal growth conditions.

### 2.2. ROS Levels Across Cold Tolerant and Susceptible Reciprocal Hybrids Agree with Cold Recovery Phenotype

Hydrogen peroxide (H_2_O_2_) was assayed as a representative ROS given its stability and involvement in both stress signaling and damage. For H_2_O_2_ content across both sets of reciprocal hybrids, the three-way interactions of hybrid, timepoint, and treatment group were significant (*p* < 0.05) sources of variation (Table 1). For cold treated CHxST8 and ST8xCH hybrids, H_2_O_2_ content was significantly (*p* < 0.05) higher in both hybrids 24 h after cold treatment compared to before or during cold treatment. For CHxMM and MMxCH, H_2_O_2_ content of MMxCH was significantly (*p* < 0.05) higher 24 h after cold treatment compared to all other hybrid-treatment combinations (Figure 3). No significant differences were observed in mean H_2_O_2_ content for either reciprocal hybrid pair in control conditions across all timepoints (Figure 3). The H_2_O_2_ levels of CHxST8 were only significantly lower than ST8xCH after cold treatment at a significance level of 10% (*p* < 0.1). 

The reduced levels of H_2_O_2_ after cold treatment for both hybrids with CH as the maternal parent (CHxMM and CHxST8) compared to their reciprocal are consistent with the cold recovery phenotype. The elevated H_2_O_2_ level in CHxST8 after cold treatment is not consistent with the results of CHxMM, suggesting that the low H_2_O_2_ level in CHxMM may be a consequence of genotype specific enhanced cold response and ROS scavenging.

### 2.3. A Maternally Transmitted SNP in the CF_1_F_O_-ATPase β-Subunit Gene (atpB) Is a Marker for Cold Tolerance

Mapped reads from sequencing of cpDNA from DHs CH, ST8, and MM had an average coverage depth of 2314, 2165, and 1056 reads at each nucleotide position, respectively. One SNP was identified in the cpDNA of cold tolerant CH compared to both cold susceptible ST8 and MM, and was located at position 56,560 in the CH chloroplast reference sequence (Table 2). This is a non-synonymous SNP within atpB that results in an amino acid change from threonine (ST8 and MM) to arginine (CH) at residue position 86 in the beta-subunit in the CF_1_ domain (CF_1_ β-subunit) of CF_1_F_O_-ATPase (βThr^86^ to βArg^86^). Additional SNPs specific to ST8 or MM compared to CH were identified at positions 59151, 122965, and 126348 (Table 2). The SNP at position 59151 was specific to ST8 and located in a non-coding region between rbcL (ribulose-bisphosphate carboxylase large subunit) and accD (acetyl-CoA carboxylase subunit). The SNP at position 122,965 was found in an underrepresented region in DH ST8 and is located in an intronic region of ndhA (NAD (P) H-quinone oxidoreductase subunit 1). The SNP at position 126,348 was only present in MM and was a synonymous mutation located in ycf1, conditioning no amino acid change due to degeneracy of codons. The ycf1 and atpB SNPs were also identified by Chung et al. [4]. 

None of the SNPs in the cpDNA were edited in the chloroplast RNA (cpRNA) across DHs and reciprocal hybrids and the SNPs in the cpRNA of reciprocal hybrids were consistent with maternal transmission of the chloroplast genome in cucumber (Table 2) [20]. At position 56,560, the SNP identified in CH cpDNA was observed in the cpRNA of reciprocal hybrids CHxST8 and CHxMM, for which CH was the maternal parent (Table 2). As expected, SNPs at positions 59,151 and 122,965 which were specific to ST8 cpDNA were also specific to ST8xCH cpRNA and the SNP at position 126348 which was specific to MM cpDNA was only identified only in MMxCH cpRNA (Table 2). Consistent with the underrepresented region found around position 122965 in ST8 cpDNA, no reads mapped to position 122,960 and low read depth (27–31 reads) mapped to positions 122,961 through 122,965. 

### 2.4. Nuclear Gene Expression Patterns Across Cold Tolerant and Susceptible Reciprocal Hybrids Agree with Cold Recovery Phenotype

Read mapping quality control and statistics indicated good coverage and alignment to the reference sequence (Appendix A). Transcriptome analysis of reciprocal hybrids across timepoints before, during, and after cold treatment revealed that the majority of differentially expressed genes (DEGs) between reciprocal hybrids were identified after cold treatment compared to before and during treatment (Figure 4). Few DEGs were found in common between both sets of reciprocal hybrids before cold treatment. Of the total of 21 DEGs, 19 genes were directionally congruent between the sets (Figure 5). Of the few DEGs identified between reciprocal hybrids, none were found in common across both reciprocal hybrid pairs at hour 3 during cold treatment (Figure 5). The sets of DEGs for MM and ST8 reciprocal crosses during cold treatment contained genes encoding an ATP binding protein and chlorophyl and DNA binding proteins, respectively. A total of 4047 genes were differentially expressed across both pairs of reciprocal hybrids after cold treatment (Figure 5). Of those genes, 1876 and 2151 were up- and down- regulated, respectively, in both the hybrids with CH as the maternal parent, and 20 genes contrasted in the direction of difference.

Gene Ontology (GO) enrichment analysis determined that there were no enriched GO terms for the DEGs in common between reciprocal sets before cold treatment. Separate evaluation of GO term overrepresentation in the two sets of DEGs indicated that both sets were enriched (*FDR* < 0.05) for genes associated with cytosolic ribosomal subunits and translation. Genes related to fatty acid metabolism, microtubules and cytoskeleton components, ATP binding, DNA binding and replication, mitotic cell cycle, and transferase activity were enriched (*FDR* < 0.05) in the set of DEGs common across both reciprocal hybrids after cold treatment (Appendix A). 

Clustering of samples indicated expression similarity between reciprocal hybrids before and during cold treatment (Figure 6). Hybrids MMxCH and ST8xCH separated from the expression pattern of the other samples and clustered together after cold treatment, indicating higher expression pattern similarity across cold susceptible hybrids than between reciprocals after cold treatment (Figure 6). The heatmap of the DEGs indicates more extreme Z-scores in the cold susceptible hybrids after cold treatment indicating more divergent expression relative to the other samples (Figure 6). 

## 3. Discussion

### 3.1. Chloroplast-Associated Cold Tolerance Does Not Reduce Growth under Normal Conditions

Reciprocal crosses of DH lines produce hybrids with identical nuclear genotypes, but the chloroplast and mitochondria are different due to differential transmission of the organellar genomes in cucumber [20]. Given its maternal transmission, the cold tolerance of CH is hypothesized to be conditioned by its chloroplast genome. DHs of MM and ST8 were considered to be cold susceptible due to significantly reduced growth 14 days after cold treatment (Figure 1 and Figure 2). Reciprocal crossing of DH CH with DHs MM and ST8 resulted in both cold tolerant (CHxMM and CHxST8) and susceptible (MMxCH and ST8xCH) hybrids with two different identical nuclear genotypes. This crossing scheme allowed us to study the genetic and physiological basis of the cold tolerance in two different nuclear backgrounds.

The maternally transmitted cold tolerance of CH was originally assessed based on damage ratings of the first true leaf directly following and up to 14 days after cold treatment [1,2,15,42]. Here we demonstrate that the maternally transmitted cold tolerance of CH shows enhanced recovery and growth after cold treatment (Figure 1 and Figure 2). While enhanced cold tolerance has been achieved by genetic engineering through overexpression of cold response transcription factors such as CBF1, reduced plant growth and yield can be an undesirable byproduct of this approach given the metabolic tax of constitutive production of metabolites and proteins [19,43]. The cold recovery phenotype conferred by CH does not impart a reduction in growth under normal conditions, indicating its potential value for enhancing cold tolerance without a metabolic drag.

### 3.2. Potential Role of the Chloroplast atpB in Conferring Cold Tolerance in CH

The non-synonymous polymorphism in atpB was the only difference conditioning an amino acid change between the chloroplast genomes of cold tolerant and susceptible DHs (Table 2). Chung et al. [4] also identified this SNP in cpDNAs of cold tolerant CH compared to susceptible cucumber lines. However, chloroplast genes can undergo post-transcriptional RNA editing with notable variability in editing sites across species [23,24]. Sequencing of the chloroplast transcriptomes of CH and susceptible cucumbers revealed no evidence of RNA editing at the SNP in atpB. 

Comparative analyses of CF_1_ β-subunit amino acid sequences across closely related *Cucumis hystrix*, other cucurbit species, and spinach (*Spinacia oleracea*) revealed high levels of conservation with a few commonly divergent residues (Figure 7). The amino acid sequence of CF_1_ β-subunit was nearly identical between CH and *C. hystrix*, except at position 86 for which its residue matched those of ST8 and MM. Although residues at amino acid position 86 of the CF_1_ β-subunit varied across other cucurbit species, only CH had an arginine at this position (Figure 7). Woessner et al. [44] also reported that the CF_1_ β-subunit amino acid sequence in *Chlamydomonas* was highly homologous to those of *Escherichia coli*, bovine heart mitochondria, and in chloroplasts of higher plants. Overall, our sequencing results indicated that there is little genetic variation in the chloroplast genome within cucumber and are in agreement with previous reports that chloroplast genomes, especially within taxa, are highly conserved in gene order and content [45,46]. Daniell et al. [47] reported significant insertions and deletions among chloroplast genomes in the Solanaceae, but observed that ATPase genes were among the least divergent.

The SNP identified in the chloroplast atpB conditions an amino acid change from threonine to arginine at position 86 of the CF_1_ β-subunit. Threonine is a polar, net neutral amino acid with a hydroxyl side chain capable of hydrogen bonds. Threonine can also function as a site of phosphorylation, essential to protein regulation and signaling [48]. Arginine is a basic amino acid with a complex guanidinium side chain, which at cellular pH is protonated and thus positively charged. The CF_1_F_O_-ATPase is a multi-subunit membrane-bound enzyme complex, made up of two domains (CF_1_ and CF_O_) that converts ADP and inorganic phosphate into ATP, driven by the electrochemical proton gradient created by light-dependent photosynthetic electron transport [49]. The CF_O_ domain is embedded in the thylakoid membrane and functions as an anchor, proton channel, and motor to drive the catalytic rotary head in the CF_1_ domain. The rotary head of the CF_1_ domain is made up of three αβ heterodimers suspended into the stroma, containing the catalytic site for ATP synthesis at the interface of the α- and β-subunits [50]. A redox mechanism on the γ-subunit of CF_1_F_O_-ATPase provides light-dependent regulation to inhibit enzymatic activity in the dark and prevent ATP hydrolysis [51].

High-resolution crystal structure of spinach CF_1_F_O_-ATPase has been resolved, allowing for protein modeling and predictions [50]. Modeling of CF_1_F_O_-ATPase complex established that this amino acid change at position 86 is outwardly facing and is located at the interface of the α- and β-subunits, but is not within the ATP synthesis catalytic region or known regulatory sites (Figure 8). Phosphorylation-dependent cross-linking and interaction, specifically at the αβ interface, has been reported and suggested to influence enzyme stability, activity, and catalytic binding [52,53]. The modification of the CF1 β-subunit in CH may therefore affect the efficiency of ATP synthesis or the regulation of the complex in response to cold stress. If this mutation constitutively inhibited the efficiency of CF_1_F_O_-ATPase in CH, it would impose a continual stress that could prime the plant prior to experiencing other stresses, but this inhibition and imposed stress would likely result in growth reduction, which was not observed in CH (Figure 1). A SNP identified in the F_1_F_O_-ATPase α-subunit gene (atpA) of cyanobacteria was found to be associated with stress tolerance to high light and high temperature [54]. This SNP was associated with elevated CF_1_ α-subunit protein levels, increased CF_1_F_O_-ATPase and photosystem II activity, and higher ATP concentrations under heat stress [54]. A similar bolstering of CF_1_F_O_-ATPase activity could also support increased cold recovery in CH. Low temperatures in light can result in photoinhibition due to oxidative damage of photosynthetic enzymes and apparatus, namely to photosystems I (PSI) and II (PSII) [55,56,57]. ATP supply has been implicated as main factor in determining the extent photoinhibition given the high ATP demand in transcription, degradation, and synthesis processes to facilitate repair [58]. Functioning of CF_1_F_O_-ATPase is essential in the supply of ATP to fuel repair processes and alleviate photodamage. Maintained ATP supply could supply sufficient energy to repair photodamaged apparatus to maintain cell function after cold. Increased CF_1_F_O_-ATPase activity and ATP content was correlated with enhanced cold tolerance in rice, resulting in better seed set and yield under cold conditions [14]. If the CF_1_F_O_-ATPase variant in CH allowed for maintained enzyme function through cold stress, this could in turn result in continued ATP supply, better recovery, and higher biomass accumulation.

Plasma membrane H+ -ATPase (PM-ATPase) has been reported to be involved in cold stress response across species, with increased activity reported after cold acclimation [59,60,61]. Short periods of low temperature resulted in reduced PM-ATPase activity in cucumber roots, but after cold treatment and in longer periods of low temperatures increased PM-ATPase activity was observed [61]. A similar trend was observed in Arabidopsis such that under cold stress at 4 °C for 6 h, PM-ATPase had reduced activity correlated with reduced association with 14-3-3 proteins, while longer exposure resulted in higher PM-ATPase activity levels and increased association with 14-3-3 proteins [60]. These findings suggest a mechanism of cold stress response and acclimation that involves regulation of PM-ATPase though altered interactions with 14-3-3 proteins [60]. 14-3-3 proteins are highly conserved, phosphopeptide-binding proteins that mediate signal transduction and protein regulation, localization, and interactions [62,63]. The CF_1_ β-subunit is a known target of protein phosphorylation [64]. 14-3-3 proteins regulate chloroplast and mitochondrial F_1_F_O_-ATPases through phosphorylation-dependent direct interaction with their respective F_1_ β-subunits [65]. Direct interaction between a leucine-rich repeat receptor-like protein kinase (LRR-RLK), CTB4a (cold tolerance at booting stage), and CF_1_ β-subunit was observed in cold tolerant rice associated with enhanced CF_1_F_O_-ATPase activity and higher ATP content [14]. Further investigation into phosphorylation and regulatory binding sites of the CF_1_ β-subunit could provide insight into CF_1_F_O_-ATPase regulation and may indicate whether protein-mediated regulation could play a role in conferring enhanced cold tolerance in CH.

### 3.3. Transcriptional Differences and ROS Levels in the Cold Recovery Phenotype

GO enrichment of DEGs common across both sets of reciprocal hybrids revealed that the cold recovery phenotype from CH may be associated with altered fatty acid metabolism, cytoskeleton structure, microtubule function, mitotic cell cycling, and transcriptional regulation (Appendix A). Alteration of fatty acid composition and metabolism has been associated with cold stress response [59,66,67]. Altering cell membrane composition and cytoskeleton structure is pivotal to maintaining stable and functional membranes and cells through cold stress and have been implicated in mediating other aspects of cold stress response [66,68,69]. In addition to their role in cell division, microtubules have also been proposed to play a role in sensing and signaling of abiotic stresses [70]. DNA replication and mitotic cell division are essential to growth, so DEGs associated with these functions may play a role in continued growth after cold treatment in the cold tolerant hybrids. Differential expression of cell cycle genes was consistent with growth reduction in cold stressed maize [71]. Additionally, DEGs involved in cold tolerance in *Brassica rapa* were associated with metabolite biosynthesis, hormone signal transduction, ribosomes, transcription regulation, and photosynthesis, and different sets of DEGs were identified at 3 and 24 h after cold treatment [72]. These similarities support the involvement of these cellular functions in enhanced cold response and recovery.

Growth reduction in MMxCH and ST8xCH after cold may also be related to the extreme gene expression response compared to their reciprocals 24 h after cold treatment (Figure 6). High numbers of common DEGs and gene expression patterns across reciprocal hybrids demonstrate divergent gene expression patterns between cold tolerant and susceptible hybrids after cold treatment, aligning with the maternally inherited cold recovery phenotype.

ROS are produced in the chloroplast and mitochondria under stress conditions and function as both a contributor to photodamage as well as a messenger for retrograde signaling and signal transduction to induce stress response [57,73,74,75]. If the CH chloroplast conferred a priming effect by inducing a continuous low level of stress response, we would expect to observe consistently elevated levels of ROS in the hybrids with CH as the maternal parent. Since there was no difference between reciprocal hybrids prior to cold treatment, there is no evidence of a priming phenotype. 

The low level of ROS in CHxMM after cold treatment indicates a reduced state of stress in this hybrid, while the elevated ROS levels in MMxCH imply higher stress response. Although early fluxes of ROS are known to trigger stress response, balancing of ROS production and removal is central to mediating stress response and damage [76,77]. The elevated ROS observed in both MMxCH and ST8xCH 24 h after cold treatment indicates an inability of these hybrids to sufficiently scavenge ROS, which can trigger cell death and result in the observed leaf damage, stunting, and even plant death [76]. The low ROS level of CHxMM compared to MMxCH after cold treatment indicates a more balanced stress response in CHxMM, which corresponds with its observed enhanced cold recovery phenotype.

H_2_O_2_-mediated retrograde signaling is central to stress response, so observed variation in H_2_O_2_ levels may influence level and type of nuclear stress response across hybrids after cold stress [78]. The absence of elevated ROS levels at the timepoint during cold treatment was unexpected. Tissue was collected at hour 3 of the 5.5 h cold treatment, which could be too late to detect ROS peaks or too early to observe ROS accumulation. Future studies measuring ROS across more timepoints during and after cold treatment could provide valuable insight into ROS dynamics to give a clearer picture of the oxidative stress experienced and ROS response across the hybrids through and after cold treatment and tie ROS signaling to the observed changes in gene expression.

## 4. Materials and Methods

### 4.1. Plant Material

Doubled haploid (DH) lines were extracted from cold tolerant CH and cold susceptible cucumbers ‘Straight 8’ (ST8) and ‘Marketmore 76’ (MM) by culturing of immature female flowers [79]. Each DH was self-pollinated and homozygosity was confirmed using 50 simple sequence repeats distributed across the seven chromosomes of cucumber [38]. Hybrids were produced by crossing the DH of CH as the female and male with DHs ST8 and MM to produce reciprocal hybrid sets with identical nuclear genotypes in two different genetic backgrounds. 

### 4.2. Cold Treatments

Seed of each reciprocal hybrid and DH parental line was sown into a soilless substrate (PRO-MIX HP Mycorrhizae; Premier Tech Horticulture, Quakertown, PA, USA) in 11.4 cm pots and germinated for 4 days on heat pads at 28 °C. After germination, plants were grown in a greenhouse at a constant temperature of 28 °C with supplemental lighting by sodium vapor lamps for a 16-h day until the second first true leaf was fully expanded and second true leaf was beginning to expand (approximately 10 to 14 days). Plants from each DH or reciprocal hybrid were randomly assigned to control or cold treatment groups. Plants assigned to the cold treatment group were moved into growth chambers at 4 °C for 5.5 h with light intensity of 270 µmol·s^−1^·m^−2^ at the top of the leaves of plants. After cold treatments, plants were moved back into the original greenhouse. Control plants remained in the greenhouse. At 1, 4, 7, and 14 days after cold treatment, subsets of five plants of each DH or reciprocal hybrid from the cold treated and control groups were harvested by cutting just above the cotyledon. Fresh weight of each plant was measured immediately after harvest using a digital balance. Harvested plants were placed in a drying oven at 60 °C for at least four days and then dry weight was measured using a digital balance. 

Data were transformed for normality due to non-constant variance. Linear fixed effect models were developed for transformed fresh and dry weight measurements, using the aov function in the stats package in R [80]. Models were developed as follows: Y_ijkl_ = µ + G_i_ + R_j_ + T_k_ + D_l_ + (GxT)_ik_ + (GxD)_il_ + (TxD)_kl_ + (GxTxD)_ikl_ + Ɛ_ijkl_, where µ is the grand mean for each Y trait; G_i_ is the effect of the *i*th hybrid genotype; R_j_ is the effect of the *j*th replication; T_k_ is the effect of the *k*th treatment; D_l_ is the effect of the *l*th day (timepoint); (GxT)_ik_ is the interaction of the *i*th hybrid genotype and *k*th treatment; (GxD)_il_ is the interaction of the *i*th hybrid genotype and *l*th day; (TxD)_kl_ is the interaction of the *k*th treatment and *l*th day; (GxTxD)_ikl_ is the three-way interaction of the *i*th hybrid genotype, *k*th treatment, and *l*th day; and Ɛ_ijkl_ is the experimental error for the *ijkl*th observation. Least-square means were calculated using the emmeans function of the emmeans package in R [81]. Reciprocal hybrids were directly compared, while each of the cold susceptible DH lines were compared to DH CH. Comparisons were made within treatment groups at each time point. Significance was determined for the models of transformed data least-square means based on Tukey’s HSD (honestly significant difference) test in the emmeans package in R [80]. Calculated least-square means calculated from models of untransformed data were plotted over time for visualization.

### 4.3. Hydrogen Peroxide (H_2_O_2_) Assays

The two sets of reciprocal hybrids (CHxST8 with ST8xCH and CHxMM with MMxCH) were used for H_2_O_2_ assays. Twenty four seeds of each hybrid were planted and germinated on hot pads at 28 °C for 4 days. Plants were then grown in a growth chamber at constant 28 °C for 10 to 12 days. Twelve plants of each hybrid were randomly selected for treatment or control groups and one of three time points (directly before cold treatment, at hour 3 during cold treatment, and 1 day after cold treatment). Treatment group plants were cold treated at 4 °C for 5.5 h as previously described, then returned to growth chamber. Control plants remained in the growth chamber. Tissue from four plants in each treatment group were harvested at each timepoint for each hybrid (4 plants × 2 treatment groups × 3 time points × 4 hybrids = 96 samples). Each plant represented a biological replicate. The apical meristem and youngest leaf was harvested and immediately frozen in liquid nitrogen. Frozen tissue samples were stored at −80 °C until processing. Each sample was then ground in liquid nitrogen using a mortar and pestle and 150 mg of finely ground tissue was diluted in 1000 µL of 1X reaction buffer and processed according to the manufacturers protocol using the Amplex Red Hydrogen Peroxide Assay Kit (Invitrogen, Carlsbad, CA, USA) in 96-well plates with a total volume of 100 μL per microplate well. Each sample was replicated four times on each plate for four technical replicates per sample. Standard dilutions (0–20 µM) of H_2_O_2_ were freshly made for each microplate assay. Absorbance was measured at 570 nm on an Infinite M1000 Pro microplate reader (Tecan, Männedorf, Switzerland). Standard curves were created using four-parameter polynomial regression and corresponding sample H_2_O_2_ concentrations were calculated using Assayfit Pro (AssayClouds, Nijmegen, The Netherlands). Total protein concentration was also measured for each sample. A 1/10 dilution of each sample was made and each diluted sample was processed according to manufacturer’s protocol using the Coomassie (Bradford) Protein Assay Kit (Thermo Fisher Scientific, Waltham, MA, USA) to assess sample protein content. Standard BSA dilutions (0–1500 µg/mL) were included on each microplate. Each protein dilution sample was replicated four times on each plate for four technical replicates. Standard curves were created using 4-parameter polynomial regression and corresponding sample protein concentrations were calculated using Assayfit Pro (AssayClouds, Nijmegen, The Netherlands). H_2_O_2_ and protein concentration measurements were averaged across the four technical replicates and H_2_O_2_ content (nmol/mg protein) was calculated for each of four biological replicates (plants).

Linear mixed effect models were developed for H_2_O_2_ measurements (nmol/mg protein) using lmer function of the lme4 package for each reciprocal hybrid pair [82]. Models included replication as a random effect and main effects of hybrid, timepoint, and treatment group, all two-way interactions of the main effects, and three-way hybrid by timepoint by treatment interaction as fixed effects. Significance of fixed and random effects were tested using the anova and ranova functions of the lmerTest package, respectively [83]. Least-square means were calculated for each hybrid, timepoint, and treatment combination and pairwise comparisons were made based on Tukey’s HSD test using the emmeans package in R [80].

### 4.4. Sequencing of Chloroplast DNA and RNA

Chloroplasts from DHs of CH, ST8, and MM were isolated using the protocol described by van Wijk et al. [84]. Plants were placed into dark for 3 days prior to chloroplast isolation. At least 60 g of leaf tissue was homogenized with 300 mL of plastid isolation buffer. Homogenate was filtered through eight layers of cheesecloth and then two layers of Miracloth. Filtrate was then centrifuged twice at 200× *g* to remove nuclei and cellular debris prior to precipitating and isolating intact chloroplasts. Chloroplast DNA was extracted from the isolated chloroplasts by a CTAB (Cetyltrimethylammonium Ammonium Bromide) protocol [85]. The cpDNA quality and concentration were determined by electrophoresis through 1% agarose gels and using the NanoDrop spectrophotometer (Thermo Fisher Scientific, Waltham, MA, USA), respectively. Samples were stored at −80 °C until shipment on dry ice to Novogene (Sacramento, CA, USA) for library construction and sequencing. Libraries were assembled using NEBNext Ultra II DNA Library Prep Kit (New England Biolabs, Ipswich, MA, USA) and then were sequenced on the Illumina platform with paired-end 150 bp (PE 150) reads. Sequences were aligned to the CH reference cpDNA [4] using the BWA software [86], and duplicates were removed by SAMTOOLS [87]. Sequences and polymorphisms were visualized using Integrative Genomics Viewer (IGV, Version 3.97, Broad Institute, Cambridge, MA, USA) [88].

Chloroplasts were isolated from approximately 40 plants of each reciprocal hybrid (CHxST8, ST8xCH, CHxMM, and MMxCH) as described above and then chloroplast RNAs (cpRNAs) were extracted. RNA extraction was performed using Direct-zol™ RNA Kits (Zymo Research, Irvine, CA, USA) according to the manufacturer’s protocol. Qualities and concentrations of cpRNAs were determined by electrophoresis through 1% agarose gels and the NanoDrop spectrophotometer (Thermo Fisher Scientific, Waltham, MA, USA), respectively. cpRNA samples from CHxST8 and ST8xCH were sent to the University of Wisconsin-Madison Biotechnology Center and samples from CHxMM and MMxCH were sent to Novogene. Libraries were assembled after rRNA depletion using Illumina Ribo-Zero Magnetic Kit and then subjected to Illumina sequencing using PE 150 reads. Sequences were aligned to the CH chloroplast reference [4] and polymorphisms visualized as described above.

### 4.5. Chloroplast Protein Modeling and Homology Comparisons

The CF_1_F_O_-ATPase structure was modeled in PyMOL (The PyMOL Molecular Graphics System, Version 2.0 Schrödinger, LLC, New York, NY, USA) to visualize the non-synonymous polymorphism in the beta-subunit in the CF_1_ domain (CF_1_ β-subunit) of CF_1_F_O_-ATPase. Previously, determined high-resolution structures of the CF_1_F_O_-ATPase complex (RCSB PDB: 6FKF, 6FKH, 6FKI) were used for modeling [50]. Amino acid sequence homology was compared across species using protein-protein BLAST (BLASTp; NCBI, Bethesda, MD, USA) and the Multiple Sequence Alignment Viewer (MSA Viewer; NCBI, Bethesda, MD, USA).

### 4.6. Nuclear mRNA Sequencing and Expression Analyses

Plants of reciprocal hybrids CHxST8, ST8xCH, CHxMM, and MMxCH were grown and cold treated as described above. The apical meristem and youngest leaf were harvested from plants directly before cold treatment, at hour 3 during cold treatment, and 1 day after cold treatment. Tissues were immediately frozen in liquid nitrogen and then ground in liquid nitrogen using a mortar and pestle. For each reciprocal hybrid, tissues from three plants were combined for each sample (biological replicate) at each timepoint (4 hybrids × 3 replicates × 3 timepoints = 36 samples). Total RNA was extracted from each sample and quality and concentration were determined as described above. Samples were stored at −80 °C until shipment on dry ice to Novogene (Sacramento, CA, USA) for library construction and mRNA sequencing. Sample mRNA was enriched using oligo(dT) beads and cDNA libraries were constructed with 250–300 bp inserts. Libraries were then sequenced on Illumina NovaSeq platform using PE 150 reads.

Raw fastq files were trimmed to remove adapter sequences using Skewer [89]. Reads were then filtered for genes with zero or low-abundance and normalized by the method of trimmed mean of M-values (TMM) [90]. Paired-end reads were aligned to the 9930 cucumber reference (GenBank accession GCA_000004075.2) using STAR (Spliced Transcripts Alignment to a Reference) [91]. Mapped reads were quantified using RSEM (RNASeq by Expectation Maximization), implementing TPM (Transcripts per million mapped reads) for accurate comparison across libraries [92]. Analysis of differentially expressed genes (DEGs) between pairs of reciprocal hybrids were performed using the glm function in the edgeR package [93]. Significance was assessed based on the adjusted *p*-value using a Benjamini-Hochberg correction to control for the false discovery rate (FDR) [94]. 

DEG sets were compared using Venny online analysis tool [95]. Gene Ontology (GO) enrichment analyses were performed for sets of differentially expressed genes using the PANTHER tools statistical overrepresentation test and significance was tested based on Fisher’s exact test with a correction for FDR [96,97,98]. Overrepresentation in gene sets was compared to a custom reference gene list which included all genes detected with at least two TTM-adjusted read counts across the experiment. 

Genes were filtered based on significant (*p* < 0.05) differential expression and log_2_(FC) ≥ 2 between at least one set of reciprocal hybrids in at least one timepoint. Average RSEM for all filtered genes was calculated across replicates for each sample group. Z-scores were calculated for each sample across all genes. Genes with divergent expression and sample expression patterns were clustered using average linkage method based on Euclidean distance and a heatmap was created and colored by Z-score using Heatmapper [99].

## 5. Conclusions

A maternally conferred, cold recovery phenotype from CH resulted in increased fresh and dry weights of cold tolerant hybrids 14 days after cold treatment. Sequencing of the cpDNA revealed a SNP in the chloroplast atpB of cold tolerant CH that was not present in cold susceptible DHs ST8 and MM. cpRNA sequencing of cold tolerant and susceptible reciprocal hybrids confirmed that the atpB SNP was maternally transmitted and maintained though transcription and RNA editing. The CH SNP results in an amino acid change in CF1 β-subunit of CF_1_F_O_-ATPase from βThr^86^ (ST8 and MM) to βArg^86^ (CH). CF_1_F_O_-ATPase activity and ATP supply are vital to oxidative stress repair and recovery. If this mutation resulted in altered CF_1_F_O_-ATPase activity or regulation through cold stress, it could allow for continued ATP supply and reduced oxidative stress. Nuclear gene expression patterns and DEGs across reciprocal hybrids supported a cold recovery phenotype with more similar gene expression response in cold tolerant hybrids after cold treatment than before or during cold. ROS levels remained at similar levels in both reciprocal hybrids before and during cold treatment, but were elevated after cold treatment for MMxCH and ST8xCH compared to their reciprocals. High ROS content in MMxCH and ST8xCH after cold may confer increased oxidative damage compared to their respective reciprocal hybrids. Patterns of gene expression and ROS levels across cold treatment were consistent with a cold recovery phenotype across both sets of reciprocal hybrids. Gene editing of chloroplast atpB in cold susceptible and tolerant hybrids and cold recovery phenotyping of the genetically edited transformants could provide evidence for the causal nature of the atpB polymorphism in conferring cold tolerance. Additional investigation of the effects of the function, regulation, potential structural changes, and protein accumulation of the variant CF1 β-subunit on the overall activity of the CH CF_1_F_O_-ATPase complex could further elucidate the mechanism of this tolerance. Confirmation of this polymorphism in imparting cold tolerance would provide a target of gene editing or selection for development of cold tolerance across cucurbits and other warm-season crops.

## Figures and Tables

**Figure 1 plants-10-01092-f001:**
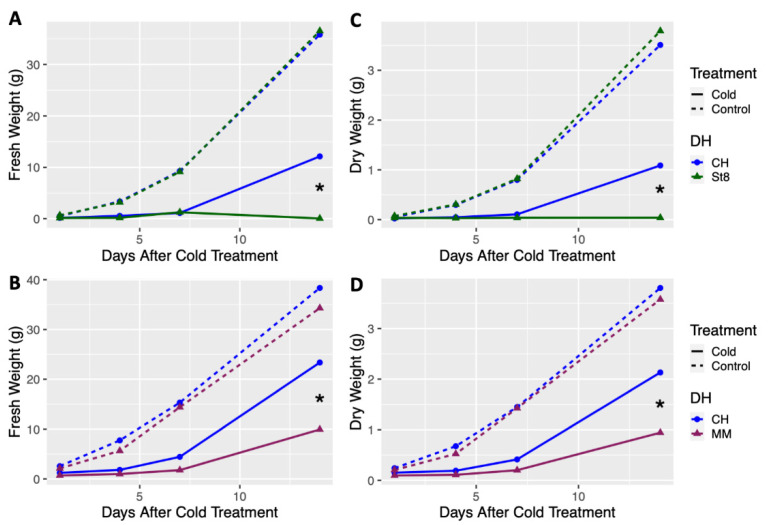
Least-square means of fresh and dry weight of doubled haploid (DH) ‘Chipper’ (CH) versus DHs (**A**,**C**) ‘Straight 8’ (St8), (**B**,**D**) ‘Marketmore 76’ (MM), plotted over timepoints at 1, 4, 7, and 14 days after cold treatment. Plants were grown in a greenhouse at 28 °C for 10 days then plants were either treated at 4 °C for 5.5 h at a light intensity of 270 µmol·s^−1^·m^−2^ (Cold) or remained in the greenhouse (Control). Plants were harvested by cutting just above the cotyledon. Each DH was compared to CH within each timepoint and treatment. * indicates significant difference between DHs based on Tukey’s HSD (*p* < 0.05).

**Figure 2 plants-10-01092-f002:**
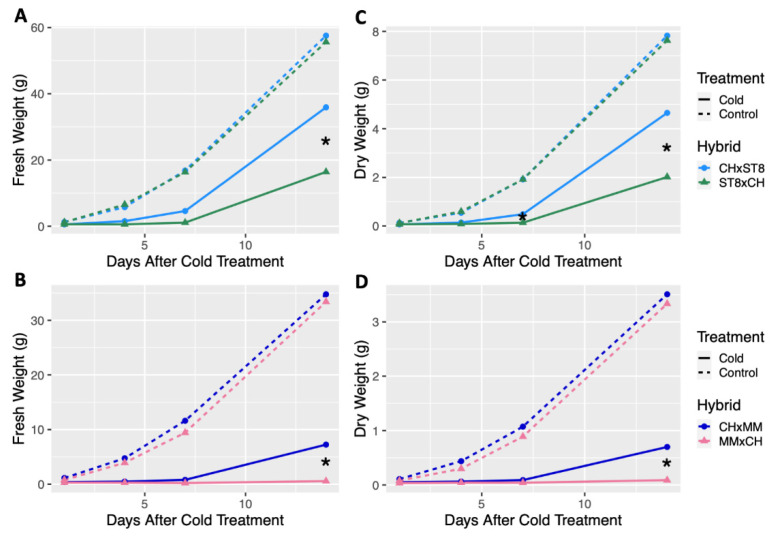
Least-square means of fresh and dry weight of reciprocal hybrids plotted over timepoints 1, 4, 7, and 14 days after cold treatment. Cold tolerant ‘Chipper’ (CH) was reciprocally crossed with (**A**,**C**) ‘Straight 8’ (ST8) and (**B**,**D**) ‘Marketmore 76’ (MM). Plants were grown in a greenhouse at 28 °C for 10 days then plants were either treated at 4 °C for 5.5 h at a light intensity of 270 µmol·s^−1^·m^−2^ (Cold) or remained in the greenhouse (Control). Plants were harvested by cutting just above the cotyledon. Hybrids were compared within each timepoint and treatment. * indicates significant difference between reciprocal hybrids based on Tukey’s HSD (*p* < 0.05).

**Figure 3 plants-10-01092-f003:**
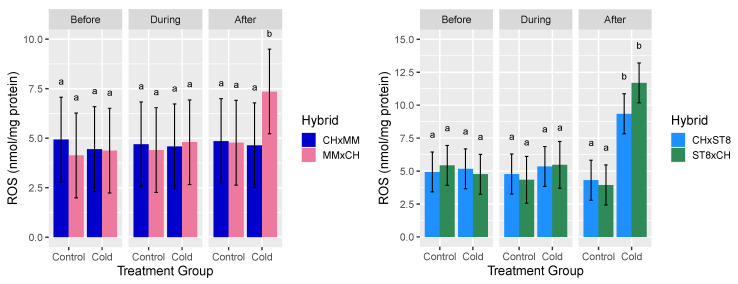
Mean reactive oxygen species (ROS) hydrogen peroxide (H_2_O_2_) level relative to total protein content (nmol/mg protein) of apical meristem and young leaf tissue of reciprocal hybrids directly before cold treatment, at hour 3 during cold treatment, and 24 h after cold treatment. Plants were grown in a greenhouse at 28 °C for 10–12 days then cold treated at 4 °C for 5.5 h at a light intensity of 270 µmol·s^−1^·m^−2^ (Cold) or remained in the greenhouse (Control). Error bars represent least-square means 95% CI and different letters indicate significantly different hybrid means across treatment combinations based on Tukey HSD test (*p* < 0.05).

**Figure 4 plants-10-01092-f004:**
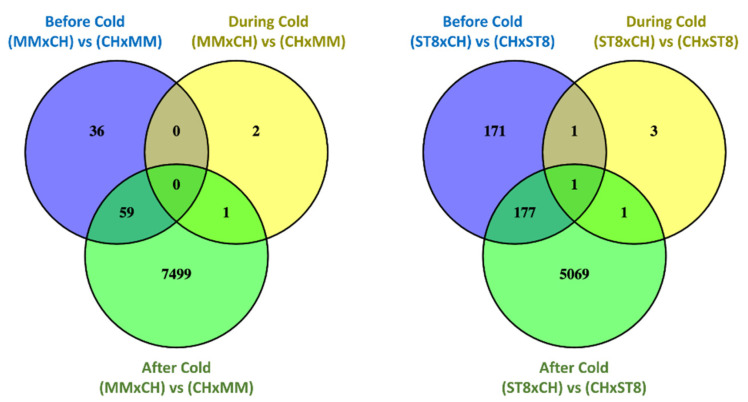
Venn diagrams comparing significantly (*FDR* < 0.05) differentially expressed genes (DEGs) between reciprocal hybrids sets, MMxCH versus (vs) CHxMM and ST8xCH vs CHxST8, before cold treatment (Before Cold; blue), at hour 3 during cold treatment (During Cold; yellow), and 24 h after onset of cold treatment (After Cold; green). Plants were grown in a greenhouse at 28 °C for 10 days then plants were treated at 4 °C for 5.5 h at a light intensity of 270 µmol·s^−1^·m^−2^.

**Figure 5 plants-10-01092-f005:**
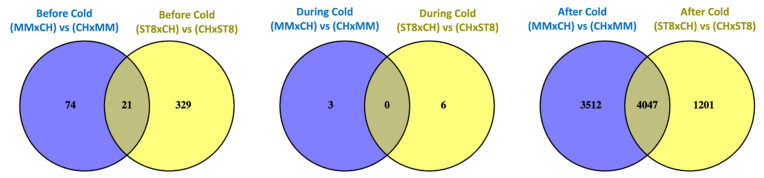
Venn diagrams comparing significantly (*FDR* < 0.05) differentially expressed genes (DEGs) across reciprocal hybrids sets, MMxCH versus (vs) CHxMM (blue) and ST8xCH vs CHxST8 (yellow), within each timepoint before cold treatment (Before Cold), at hour 3 during cold treatment (During Cold), and 24 h after onset of cold treatment (After Cold). Plants were grown in a greenhouse at 28 °C for 10 days then plants were treated at 4 °C for 5.5 h at a light intensity of 270 µmol·s^−1^·m^−2^.

**Figure 6 plants-10-01092-f006:**
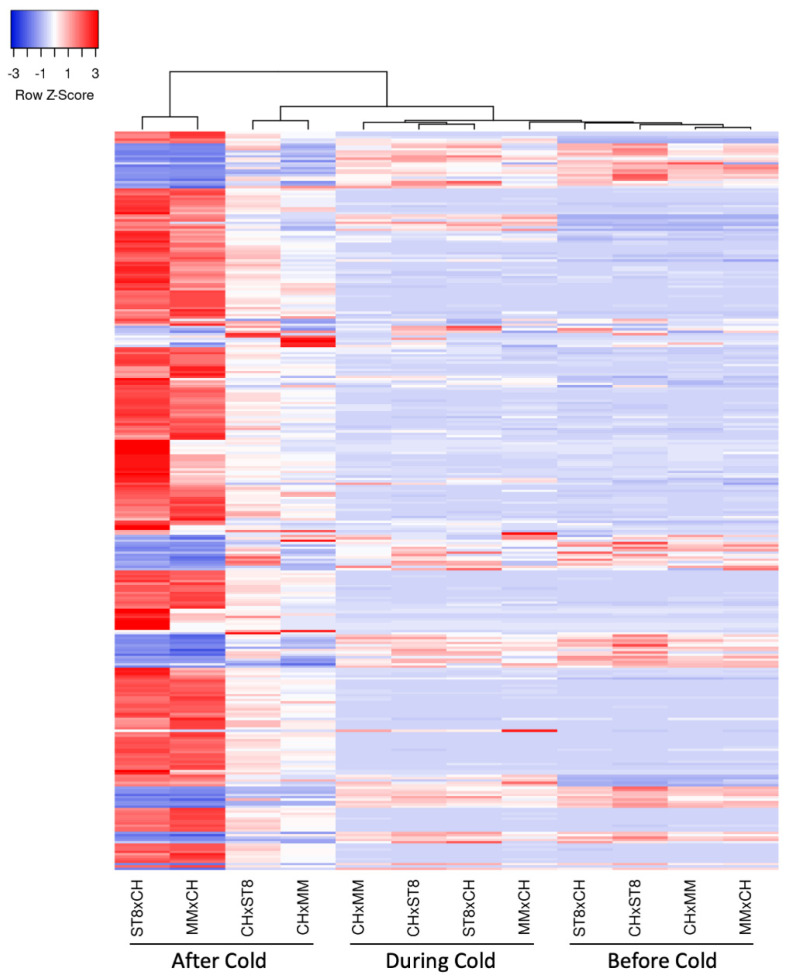
Heatmap of relative expression of significantly (*FDR* < 0.05) differentially expressed genes (DEGs; rows) with a log2FC > 2 in at least one timepoint, colored by Z-score calculated for each gene across samples (columns). Genes and samples are clustered using average linkage method based on Euclidean distance. Hybrids (CHxMM, MMxCH, CHxST8, and ST8xCH) were grown in a greenhouse at 28 °C for 10 days then cold treated at 4 °C for 5.5 h at a light intensity of 270 µmol·s^−1^·m^−2^. Samples were collected before cold treatment (Before Cold), at hour 3 during cold treatment (During Cold), or 1 day after onset of cold treatment (After Cold).

**Figure 7 plants-10-01092-f007:**
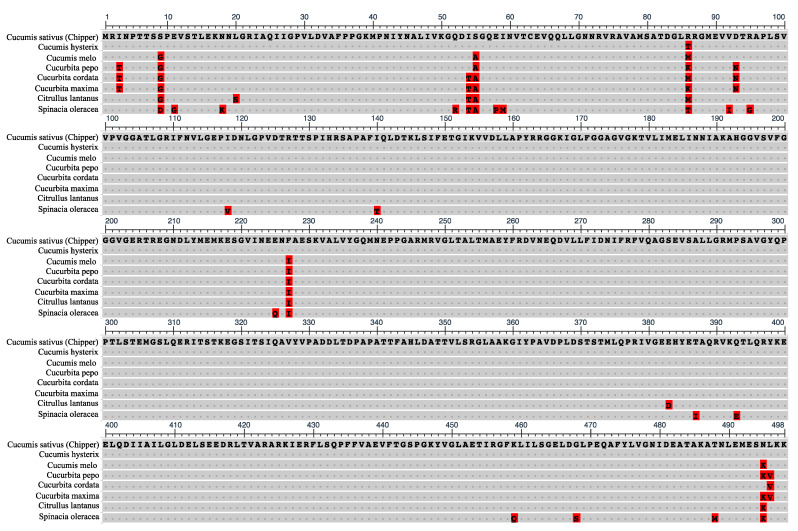
Alignment of the amino acid sequence of the ‘Chipper’ chloroplast F_1_F_O_-ATP synthase β-subunit gene with other plant species. Dots represent homology with ‘Chipper’ sequence. Highlighted amino acids indicate polymorphisms. Unique Chipper polymorphism is located at position 86.

**Figure 8 plants-10-01092-f008:**
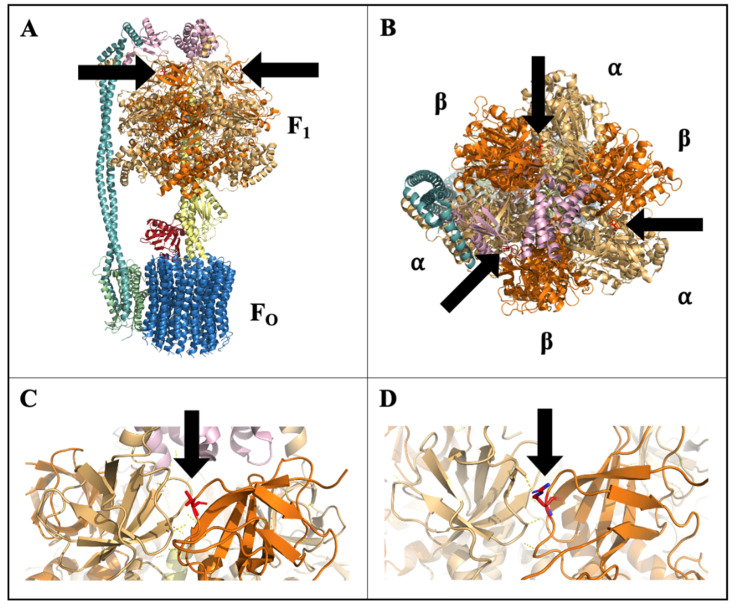
Modeled structure of chloroplast F_1_F_O_-ATP synthase (CF_1_F_O_-ATPase) with stick-structure of the amino acid at residue position 86 of the CF_1_ β-subunit (βThr^86^) shown, colored in red, and denoted by arrows: (**A**) side-view of full CF_1_F_O_-ATPase structure, (**B**) top-view showing βThr^86^ at the αβ interface, (**C**) close-up of wild-type βThr^86^ at αβ interface, and (**D**) close-up of ‘Chipper’ amino acid change to βArg^86^ at αβ interface.

**Table 1 plants-10-01092-t001:** Sources of variance and significance from analysis of variance (ANOVA) for fixed effects in mixed effect model of ROS measurements (H_2_O_2_ nmol/mg protein) for the two reciprocal hybrid pairs (CH crossed with ST8 and MM) with two treatment groups (cold and control) over three timepoints (before, during, and after cold treatment). *, **, and *** indicate significance at *p* < 0.05, 0.01, and 0.001, respectively.

Source	Mean Sq	DF	F Value	Prob > F	
*ST8xCH and CHxST8*					
Hybrid	0.98	1	1.00	0.324	
Time	27.24	2	27.87	<0.001	***
Treatment	62.24	1	63.67	<0.001	***
HybridTime	1.42	2	1.45	0.249	
Hybrid × Treatment	1.74	1	1.78	0.191	
Time × Treatment	49.59	2	50.73	<0.001	***
Hybrid × Time × Treatment	3.30	2	3.37	0.046	*
*MMxCH and CHxMM*					
Hybrid	0.92	1	2.29	0.139	
Time	4.05	2	10.08	<0.001	***
Treatment	1.96	1	4.89	0.034	*
Hybrid × Time	3.37	2	8.40	0.001	**
Hybrid × Treatment	5.41	1	13.46	<0.001	***
Time × Treatment	1.91	2	4.75	0.015	*
Hybrid × Time × Treatment	1.60	2	3.98	0.028	*

**Table 2 plants-10-01092-t002:** Single nucleotide polymorphisms (SNPs) identified across the chloroplast DNAs (cpDNAs) of doubled haploid (DH) lines ‘Chipper’ (CH), ‘Straight 8’ (ST8), and ‘Marketmore 76’ (MM) and across chloroplast RNAs (cpRNAs) of reciprocal hybrids CHxST8, ST8xCH, CHxMM, and MMxCH.

		Base Pair Position ^a^			
**Sample**		56560	59151	122965	126348
**cpDNA**	CH	C	A	C	C
	ST8	G	C	T^b^	C
	MM	G	A	C	T
**cpRNA**	CHxST8	C	A	C	C
	ST8xCH	G	C	T ^b^	C
	CHxMM	C	A	C	C
	MMxCH	G	A	C	T

^a^ Position relative to ‘Chipper’ chloroplast reference genome. ^b^ Identified within an underrepresented region.

## Data Availability

The data generated in this study are publicly available in the NCBI’s Sequence Read Archive (SRA) under accession number PRJNA728122.

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
