# Peer review of "Polymorphism in the Chloroplast ATP Synthase Beta-Subunit Is Associated with a Maternally Inherited Enhanced Cold Recovery in Cucumber"

_plants, 2021, doi:10.3390/plants10061092_

Round 1

Reviewer 1 Report

The manuscript by Oravec and Havey describes the genetic analysis of a cold-tolerant cucumber cultivar, termed “Chipper” (CH). Their study identified a mutation within the chloroplast atpB gene of CH, which might be responsible for the cold-resistant phenotype. Further transcriptome analyses were performed comparing different crosses between CH and two cold-sensitive doubled haploid lines in addition to assays about ROS accumulation and growth changes at room temperature and cold.

In general, the manuscript is interesting and data about altered chloroplast ATP synthetase responsible for cold tolerance would be a fascinating dataset for understanding cold tolerance. However, I would propose to work on the data representation, especially concerning the transcriptomic data, to make this manuscript suited for a broader audience which is desirable prior to publication.

Major points:

The study is rather difficult to follow caused by the extensive use of abbreviations of the different lines (e.g. CHxMM) or MMxCHvCHxMM (in Figure 4). The authors should work on improving the readability in that respect. It would be also good to introduce the DHs ST8 and MM76 at the beginning of the respective sections of the results. How were they generated, what do they represent, etc.

From the results and the discussion chapter, it is not quite clear what we can learn from the transcriptional analyses. There is a strong focus on the comparison of the different crosses. This section is extremely difficult to understand for readers without a genetics background. Instead of the intensive comparison of MMxCH vs CHxMM etc, it might be more interesting to really focus on the DEGs that are altered in the CH line (i.e. under cold) and how this could be linked to a mutated cpCF1F0 ATP synthetase. I am not sure if the Venn diagrams (Figure 4/5) are so informative to justify 2 main Figures. Rather highlight the DEGs of CH for RT vs cold.

At the current stage the manuscript it is not fully convincing in terms that the mutation in atpB is solely responsible for the improved cold-tolerance of CH. It would be important to show that the mutation also influences atpB expression (this might be seen in the transcript data) and protein level – consistent with the findings in other systems as stated in the discussion.

Minor points:

Figure 1 and 2:

State in the legends which part of the plant material was used for determining FW (it is given in Methods but it helps understanding the graph).

The symbols are not very intuitive, maybe use same color for one condition and symbols for different lines.

Figure 6:

It would be nice to better visualize the conditions instead of letters before hybrid names. B, C, A are very cryptic labels.

Section 2.3. Visualize identified SNPs in a cartoon, showing the position within the gene. The table is difficult to read.

Reviewer 2 Report

In this paper, the authors investigate the maternal inheritance of cold tolerance in Chipper cultivar of cucumber and identify a SNP in the chloroplast ATP synthase b-subunit that may be responsible of it.  Furthermore, they study ROS production in different sets of reciprocal hybrids and analyse the transcriptional differences between them. The work is appropriately designed and well performed using a variety of techniques, and report data an conclusions that can be interesting for the scientific community.

Therefore, I recommended accepting it for publication.

Round 2

Reviewer 1 Report

I have no further suggestions.